# NXP031 Improves Cognitive Impairment in a Chronic Cerebral Hypoperfusion-Induced Vascular Dementia Rat Model through Nrf2 Signaling

**DOI:** 10.3390/ijms22126285

**Published:** 2021-06-11

**Authors:** Jae-Min Lee, Joo-Hee Lee, Min-Kyung Song, Youn-Jung Kim

**Affiliations:** 1College of Nursing Science, Kyung Hee University, Seoul 02447, Korea; sunjaesa@hanmail.net; 2Department of Nursing, Graduate School, Kyung Hee University, Seoul 02447, Korea; lovejjoo2@naver.com; 3Robert Wood Johnson Medical School Institute for Neurological Therapeutics, Rutgers Biomedical and Health Sciences, Piscataway, NJ 08854, USA; ms3068@rwjms.rutgers.edu

**Keywords:** vascular dementia, cognition, oxidative stress, blood-brain barrier, DNA aptamer, ascorbic acid

## Abstract

Vascular dementia (VaD) is a progressive cognitive impairment caused by a reduced blood supply to the brain. Chronic cerebral hypoperfusion (CCH) is one cause of VaD; it induces oxidative stress, neuroinflammation, and blood-brain barrier (BBB) disruption, damaging several brain regions. Vitamin C plays a vital role in preventing oxidative stress-related diseases induced by reactive oxygen species, but it is easily oxidized and loses its antioxidant activity. To overcome this weakness, we have developed a vitamin C/DNA aptamer complex (NXP031) that increases vitamin C’s antioxidant efficacy. Aptamers are short single-stranded nucleic acid polymers (DNA or RNA) that can interact with their corresponding target with high affinity. We established an animal model of VaD by permanent bilateral common carotid artery occlusion (BCCAO) in 12 week old Wistar rats. Twelve weeks after BCCAO, we injected NXP031 into the rats intraperitoneally for two weeks at moderate (200 mg/4 mg/kg) and high concentrations (200 mg/20 mg/kg). NXP031 administration alleviates cognitive impairment, microglial activity, and oxidative stress after CCH. NXP031 increased the expression of basal lamina (laminin), endothelial cell (RECA-1, PECAM-1), and pericyte (PDGFRβ); these markers maintain the BBB integrity. We found that NXP031 administration activated the Nrf2-ARE pathway and increased the expression of SOD-1 and GSTO1/2. These results suggest that this new aptamer complex, NXP031, could be a therapeutic intervention in CCH-induced VaD.

## 1. Introduction

Vascular dementia (VaD) is the second most common dementia after Alzheimer’s disease (AD). It is caused by chronic cerebral hypoperfusion (CCH), leading to multiple microinfarcts and vascular cognitive impairment. CCH-induced VaD damages neurons in vulnerable areas, leading to oxidative stress and inflammation. Various animal models, such as vessel occlusion, multiple infarcts, and thromboembolism, have been developed to study VaD. Most VaD research studies use the permanent bilateral common carotid occlusion (BCCAO) model [1]. In rats, the BCCAO model has been widely used to study CCH, which induces memory deficits and neuronal cell death in the brain [2]. An increased level of reactive oxygen species (ROS) in the neurovascular unit leading to blood-brain barrier (BBB) disruption is associated with VaD [3,4]. The BBB is a highly selective permeability barrier, primarily formed of highly specialized endothelial cells containing complex tight junctions proteins, including occludin, claudin-5, the zonula occludens-1 (ZO-1), astrocyte end-feet, pericytes, and the basal lamina [5]. Oxidative stress causes BBB dysfunction [6]. ROS-related pathways that trigger BBB dysfunction include excitotoxicity, mitochondrial dysfunction, microglial activation, tight junction modification, matrix metalloproteinases activation, and inflammation activation [7]. BBB degradation may play an essential role in CCH-induced cognitive impairment [3].

Vitamin C, also known as ascorbic acid, is an essential nutrient present in various foods and sold as a dietary supplement. It is used to prevent and treat tissue damage and is particularly effective at reducing oxidative damage [8]. A ubiquitous water-soluble antioxidant and a cofactor for several enzymes, ascorbic acid can inhibit ROS generation and directly scavenge oxygen and nitrogen-based radical species generated during normal cellular metabolism [9]. In neurons, millimolar concentrations of ascorbic acid effectively protect against several causes of apoptotic neurodegeneration [10] and oxidative stress (which would otherwise lead to massive glutamate release and subsequent excitotoxicity) [11]. The brain is the organ with the highest vitamin C concentration. Plasmatic ascorbic acid slowly crosses the BBB through the glucose transporter GLUT1. Ascorbic acid also enters the brain via a two-step mechanism: first, the sodium-dependent transporter-2 (SVCT-2) transports it into the cerebrospinal fluid, and then GLUT1 transporters transfer it into the neurons [12,13]. A population study has shown that vitamin C supplements protect against VaD and cognitive impairment [14].

With their relatively small size, aptamers, produced by chemical synthesis and versatile chemical modification, are highly stable. They are emerging as an important source of new therapeutic molecules [15]. As previously reported, we developed NXP031, a DNA aptamer that binds explicitly to vitamin C, inhibits its oxidation by several oxidizing agents, and maintains this antioxidant activity during long-term storage [16].

Nuclear factor-erythroid 2-related factor 2 (Nrf2) is a master regulator that induces the expression of several cytoprotective factors, such as antioxidative enzymes and anti-inflammatory and transcription factors. The induction occurs by activating the antioxidant response element (ARE) in the promoter region of Nrf2 target genes. The Nrf2 target ARE-driven genes encode proteins that regulate a wide range of biological processes and mediate the expression of antioxidant enzymes [17]. Activation and nuclear accumulation of Nrf2 upregulate endogenous antioxidant defenses, restoring cellular redox homeostasis. Several recent studies have suggested that the Nrf2 pathway is a therapeutic target for brain injury and oxidative stress after ischemic stroke [18,19].

The present study aims to evaluate the effect of NXP031 on cognitive impairment and BBB disruption regarding antioxidant mechanisms in the CCH-induced VaD rat model.

## 2. Results

### 2.1. NXP031 Alleviates CCH-Induced Cognitive Impairment

We evaluated the effect of NXP031 on CCH-induced cognitive impairment using the novel object recognition (NOR), radial 8-arm maze, and passive avoidance tests. To confirm the difference between NXP031 concentrations, the mixture was mixed in two combinations. In this experiment, 200 mg/4 mg/kg of ascorbic acid:aptamin was used as a medium concentration, and 200 mg/20 mg/kg was used as a high concentration. The cognitive function tests showed that the NXP031 group had better spatial learning, recognition function, and short-term memory than the VaD (vehicle) group. Figure 1A presents the NOR test results. The NOR allows verifying cognitive function by assessing habituation, exploration, and new recognition periods, assessed as the difference in search times for new or familiar objects. The VaD group had a lower discrimination index (DI) in the NOR test than the sham group. However, the VaD + NXP031_M and _H groups had a significantly higher DI than the VaD group (F = 25.34, *p* < 0.001). Figure 1B shows the radial 8-arm maze test results. Both of the VaD + NXP031 groups had a significantly higher number of correct choices than the VaD group, showing better spatial learning and memory function (F = 22.02, *p* < 0.001). In the passive avoidance test (which assesses short-term memory), the VaD + NXP031 groups also had significantly better results than the VaD group (Figure 1C) (F = 38.88, *p* < 0.001), and there was no difference between the NXP031_M and _H groups. Overall, these results indicated that NXP031 attenuated CCH-induced cognitive impairment. However, vitamin C administered alone did not affect cognitive function, so we excluded the vitamin C group in all subsequent experiments.

### 2.2. NXP031 Inhibits CCH-Induced Microglial Activation in the Hippocampus

We evaluated the effect of NXP031 on CCH-induced neuronal cell death in the hippocampus through immunohistochemistry using the neuronal marker NeuN. Spatial learning and memory are hippocampal-dependent cognitive functions involving the hippocampal CA1 and CA3 pyramidal neurons. Figure 2A shows representative photomicrographs of NeuN immunostaining in the hippocampal CA1 and CA3 regions for all groups. The sham and VaD groups had similar NeuN-positive cell counts in the hippocampal CA1 and CA3 regions. We evaluated the effect of NXP031 on CCH-induced microglial activation in the hippocampus through immunohistochemistry with the microglial marker Iba-1. CCH significantly increased Iba-1 expression in the hippocampus. We measured the area percentage of Iba-1-expressing microglia in the hippocampal CA1 region. NXP031 reduced the CCH-induced Iba-1-expressing microglia (Figure 2B).

### 2.3. NXP031 Alleviates CCH-Induced Microvessel Damage and BBB Disruption

#### 2.3.1. NXP031 Decreases CCH-Induced Damage to Microvessels in the Hippocampus

Figure 3A,B show the CCH-induced narrow, short, and irregular patterns in the hippocampus microvessels. Using RECA-1 as an endothelial cell marker, we counted the microvessels that were shorter and longer than 10 μm in a 200 μm × 200 μm area. The VaD group had more fragmented and shorter microvessels than the sham group. However, the VaD + NXP031 groups had less fragmented (F = 16.54, *p* < 0.001) and longer microvessels than the VaD group (F = 12.92, *p* < 0.001). These results show that NXP031 alleviated CCH-induced damage to microvessels. In the sham group, the PDGFRβ-labeled pericytes surrounded the endothelial cells, and their morphology was also intact (Figure 3A). The CCH-induced damage seemed to contribute to pericyte loss and microvessels fragmentation directly. However, NXP031 decreased microvessels damage.

#### 2.3.2. NXP031 Reduces the BBB Against CCH-Induced Damage

The BBB is a neurovascular unit component. It includes endothelial cells, neurons, astrocyte end-feet, pericyte, microglia, and basement membranes. To evaluate its integrity, we identified endothelial cells (RECA-1), platelet endothelial cell adhesion molecules (PECAM-1), basal membranes (laminin), and pericytes (PDGFRβ). Figure 3C shows that CCH-induced VaD significantly reduces PDGFRβ expression (F = 77.01, *p* < 0.001). However, NXP031 increased the PDGFRβ consistent with the IF in Figure 3A. The VaD group also had significantly lower laminin (F = 35.41, *p* < 0.001) and PECAM-1 (F = 23.26, *p* < 0.001) expression levels than all of the VaD + NXP031 groups. NXP031 restored microvessels by decreasing CCH-induced BBB disruption.

### 2.4. NXP031 Upregulates Nrf2 Expression in the Hippocampus

Nrf2 is the master switch controlling the cellular redox status; it controls the transcription of various antioxidative genes. Figure 4A represents the IF and WB results regarding Nrf2 and keap1 signaling. In the VaD group, Nrf2 was located primarily in the cytosol, and CCH reduced its expression (F = 102.67, *p* < 0.001). In contrast, NXP031 increased Nrf2 expression and distributed it evenly in the nucleus and cytoplasm. There was no difference in keap1 expression between groups. To confirm the effect of the Nrf2 signaling on antioxidant enzymes, we assessed SOD-1 and GSTO1/2 expression levels by WB. As a result, both GSTO1/2 (F = 100.60, *p* < 0.001) and SOD-1 (F = 137.07, *p* < 0.001) expression levels were significantly higher in the NXP031-treated groups than in the VaD group (Figure 4B). Besides, the VaD + NXP031_H group had significantly higher expression levels than the VaD + NXP031_M group. Thus, Nrf2 signaling increases antioxidative enzyme expression, which has a neuroprotective effect.

### 2.5. NXP031 Suppresses CCH-Induced 4HNE Expression in the Hippocampus

4HNE accumulates in numerous oxidative stress injuries and related diseases such as neurodegenerative diseases. 4HNE is a toxic by-product of lipid peroxidation. CCH significantly increases 4HNE immunoreactivity in the hippocampus CA3 region (F = 26.08, *p* < 0.001). In NXP031 groups, the CCH-induced 4HNE expression increase was suppressed significantly in the hippocampal CA3 region (Figure 5).

## 3. Discussion

In this study, we found a new DNA aptamer/vitamin C complex, NXP031, ameliorated against CCH-induced cognitive impairment and BBB disruption by increasing antioxidant enzyme activity through potentiation of the Nrf2-ARE pathway and reducing CCH-induced ROS production. VaD severely impairs the executive function of memory [1] and accounts for more than 20% of all dementia cases, making it the second highest in proportion after AD [20]. According to clinical–pathological studies, mixed pathologies including ischemic lesions, amyloid plaque, and neurofibrillary tangles, which are characteristic of AD, account for the most significant proportion [21]. These findings suggest that understanding the mechanism of cognitive impairment caused by vascular problems can help to target and regulate vascular risk factors to slow disease progression.

In this study, we did not directly measure changes in CCH-induced brain blood flow. However, through the behavioral experiment (radial 8-arm maze test), we found that cognitive deficits occurred 11 weeks after BCCAO surgery (Appendix A). Microglia are immune cells that reside in the central nervous system (CNS) and generally play an important role in immune surveillance and maintenance of homeostasis [22]. Microglia activation causes cognitive impairment in the rotenone-induced Parkinson’s disease (PD) model [23], post-ischemic injury [24], and post-BCCAO [25] through neuroinflammation, oxidative stress, and apoptosis. As in previous studies, CCH did not cause neuronal loss, but microglia activation occurred in the VaD group [26]. Cerebral microvessels continuously deform during aging and neurodegenerative diseases, such as VaD, AD, and PD [27]. In our study, hippocampal microvessels’ fragmentation due to CCH increased significantly in the VaD group, whereas it decreased in the NXP031 group, which indicates that NXP031 prevented microvascular construction changes (Figure 3). The BBB is a highly specialized brain endothelial structure in the central nervous system. It effectively prevents harmful substances and large molecules from transferring from the peripheral blood to the brain parenchymal tissue. The BBB forms a physical barrier by junctional proteins between endothelial cells, mainly classified as tight junctions and adhesive junctions [28]. Dysfunction of tight junctions and adhesive junctions in cerebral hypoxia or oxidative stress contributes to increasing paracellular permeability [29]. Pericytes and endothelial cells are linked to the shared basement membrane by several integrin molecules and contribute to BBB maintenance [30]. In our study, the VaD group had low laminin, PECAM-1, and PDGFRβ expression levels, but NXP031 administration restored their expression. These results show that while chronic hypoperfusion directly damages the BBB, the center of cerebral homeostasis, NXP031 administration alleviated the BBB disruption.

The brain is rich in polyunsaturated fatty acids, consumes large amounts of oxygen (20% of the body consumption), and requires large amounts of ATP to maintain homeostasis, so it is susceptible to oxidative damage. The brain consumes a quarter of its energy to maintain the lipid membrane damaged by ROS [31]. Several studies have demonstrated that blocking lipid peroxide production can help prevent the development of neurodegenerative diseases [32,33]. The mechanisms by which ROS damages cells include peroxidation of polyunsaturated acids in cell membranes, DNA mutations, and nitration and carbonylation of proteins and lipids. The main products of peroxidation are the 3-carbon dialdehyde species malondialdehyde (MDA) and 4HNE [34]. MDA is the most mutagenic lipid peroxidation product, and 4HNE is regarded as the biomarker of oxidative stress in the hippocampus [35]. 4HNE was elevated in AD patients and associated with cognitive impairment [36]. It is also significantly elevated in the MPTP-induced PD model [37] and MCAO model [38]. However, there are few reports of changes in 4HNE in CCH after BCCAO. In our study, 4HNE-positive cell count in the hippocampal CA3 region increased in the VaD group and decreased significantly in the NXP031 groups. In order to evaluate the antioxidant effect of NXP031, the activity of Nrf2 was investigated.

Nrf2, the transcription factor, is one of the major regulators of antioxidant defense and plays an indispensable role in regulating the expression of antioxidant genes [39]. Under normal conditions, Keap1 maintains Nrf2 in an inactive state in the cytoplasm by continuously promoting its ubiquitination [40]. Under oxidative stress conditions, Nrf2 is released from the Keap1-Nrf2 complex and translocates from the cytoplasm to the nucleus, where it binds to the AREs [41]. AREs appear in the promoters of many antioxidant and detoxifying enzyme genes, including HO-1, SOD, GST, and NQO1 [39]. Several studies have demonstrated that increasing the activity of the Nrf2 pathway in animal models of ischemic stroke can protect against brain injuries [42,43]. Our results indicate that NXP031 treatment-induced Nrf2 expression and translocation into the nucleus under CCH-induced oxidative stress conditions (Figure 4A). CCH decreased Nrf2 expression, and NXP031 treatment increased it. A previous study reported that BCCAO increased or activated Nrf2 expression [44]. However, the present study showed that BCCAO significantly reduced Nrf2 expression levels in the hippocampus after 14 weeks. Several studies have found that CCH injury significantly reduced Nrf2 mRNA and protein levels in the hippocampus [45,46]. Recent studies also showed that a decrease in Nrf2 and SOD-1 levels in the hippocampus occurred in a CCH model [47,48,49]. Activation of the Nrf2 signaling pathway increases endogenous antioxidant SOD activity and HO-1 levels in the hippocampus [50]. We found that NXP031 treatment upregulated antioxidant SOD-1 activity and GSTO1/2 levels by upregulating Nrf2 expression in the hippocampus compared with the vehicle treatment (VaD group).

Vitamin C is a primary antioxidant that mediates several beneficial effects on the immune system, inflammatory aging, redox pathways, and mitochondrial pathways [51]. Many mammalian species synthesize it from glucose in the liver; humans only absorb it through their diet [52]. The physiologically beneficial action of vitamin C comes from its ability to donate electrons as a reducing agent. In its solid form, vitamin C is stable, but it quickly decomposes when dissolved in water. It oxidizes easily by external factors such as air, moisture, or light [53]. A DNA aptamer that prevents oxidation of vitamin C and maximizes its effectiveness increased the half-life of vitamin C by 1.7-fold in certain commercially available vitamin water formulations [54]. In addition, vitamin C reduced oxidation to less than the half-life of 2 weeks, but this DNA aptamer/vitamin C complex maintained high binding affinity for an extended period (12 weeks) [55]. In addition to previous studies reporting neuroprotective effects in PD models [16], our research results demonstrate that the cerebrovascular integrity strengthening and neuroprotective effect of NXP031 protects from CCH-induced oxidative stress and cognitive deficit. This mechanism may strengthen the antioxidant defense system by activating the Nrf2-ARE signaling pathway, thereby maintaining the BBB’s integrity. This study suggests that the novel neuroprotective NXP031 is a promising treatment against cognitive impairments such as MCI and VaD.

## 4. Materials and Methods

### 4.1. Animals

Eight-week-old male Wistar rats (180 ± 20 g) were purchased for the present study (Orient Bio, Seongnam, Gyeonggi-do, Korea). Animals were housed in a place where water and food were freely consumed, and the environment was regulated (22 ± 2 °C, humidity of 50%), and day and night cycle (12 h light/12 h dark) controlled conditions. This study conducted experiments under the National Institutes of Health Guide for the Care and Use of Laboratory Animals and approval of the Institutional Animal Care and Use Committee (IACUC) of the Kyung Hee University (KHSASP-19-080).

### 4.2. Bilateral Common Carotid Arteries Occlusion (BCCAO) Procedure

Prior to surgery, the rats (body weight 250–350 g, 12 weeks old) were anesthetized with 2% isoflurane in 70% N_2_O and a balance of O_2_. The bilateral common carotid artery occlusions were performed to avoid damage to the vagus nerve and surrounding tissues. Both carotid arteries were exposed by a ventral midline incision and double-ligated with 3-0 silk (Ailee, Korea) immediately below the carotid bifurcation. Sham animals underwent the same operation procedure without vessel ligation.

### 4.3. NXP031 Preparation

The aptamer is a single-stranded DNA that can selectively bind to a target molecule and is isolated by the SELEX method [54] and was provided by Nexmos Co., Ltd. The purified DNA aptamer was dissolved in 1 mM MgCl_2_ PBS at 95 °C for 5 min and then cooled to room temperature to form a tertiary structure. L-Ascorbic acid (Sigma-Aldrich, St. Louis, MO, USA) was added at a ratio of 1:50 (*w/w*) and then adjusted to pH 5.7 to form the final NXP031. To confirm the difference between NXP031 concentrations, the mixture was mixed in two combinations. In this experiment, 200 mg/4 mg/kg of ascorbic acid:aptamin was used as a medium concentration, and 200 mg/20 mg/kg was used as a high concentration.

### 4.4. Experimental Design

Male Wistar rats (12 weeks of age) were randomly divided into four groups (n = 10 in each group): the sham group (vehicle), the VaD group (vehicle), VaD + NXP031 medium concentration (NXP031_M), and VaD + high concentration (NXP031_H). BCCAO procedure was performed at 12 weeks of age. Twelve weeks after BCCAO, a radial 8-arm maze test was performed to confirm cognitive impairment induced by CCH. NXP031 groups were given treatment for 2 weeks simultaneously every day. Cognitive behavioral experiments were conducted for 2 weeks after treatment was terminated. All rats were sacrificed the next day after the end of all behavioral tests.

### 4.5. Behavior Tests

#### 4.5.1. Novel Object Recognition Test

12 weeks after BCCAO, the novel object recognition (NOR) test was used to evaluate cognitive function in animals. On the day before the NOR test, rats were allowed to explore the empty box (60 × 60 × 40 cm) freely for 10 min. On the first day of the test, two identical objects (T_F_ and T_F_) were placed in the upper right and lower left quadrants of the box, and rats were allowed to explore the objects in the box for 10 min. On the second day of the test, two different objects (T_N_ and T_F_) were placed, where object F was the same object used in the familiarization session, and object N (with T_N_ being dissimilar to T_F_) was the new object. The rats were then put back into the box for a 10 min period of exploration. The behavior of the rats was video recorded and the exploration of the rats was analyzed to measure the time spent exploring each object. Interest was based on the distance ≤ 1 cm between the nose and the object. The evaluation of NOR memory was expressed as a percentage of the discrimination index (DI) calculated according to the following formula:Discrimination Index = T_N_/T_N_ + T_F_
T_N_: time exploring novel object
T_F_: time exploring familiar object

#### 4.5.2. Radial 8-Arm Maze Test

The radial 8-arm maze test was performed to measure working memory. The radial 8-arm maze consisted of eight equally spaced arms radiating from a central octagonal platform. The rats were placed on the central octagonal platform, following which they were required to collect hidden water placed at the ends of each arm during a period of 8 min. Since there was water at the end of each arm, its contents were not visible from the central platform. An arm choice was scored if the rat crossed the halfway line with all four paws down an arm. Rats were allowed to enter the arms in any order until all arms had been visited or 8 min had elapsed. Each arm entry was recorded, and re-entry into previously visited arms was recorded as an error. We calculated the number of correct choices made before the first error. The test was terminated either when the rat had entered all the arms or after 8 min.

#### 4.5.3. Passive Avoidance Test

The passive avoidance test is used to evaluate learning and memory in rodent models. The passive avoidance test apparatus was divided into two zones, a bright and dark chamber with lighting, and the floor was made of wire mesh. The rats were allowed to acclimate for 60 s without the lights on in an illuminated chamber and then the lights were turned on. As soon as the rat moved to the dark chamber, an electric shock was applied for 0.5 mA for 2 s. The memory test was conducted 24 h after the training session. The latency times for the rat to enter the dark chamber were measured for 300 s.

### 4.6. Tissue Preparation

Rats were anesthetized by inhalation with ether, perfused with 0.01 M phosphate buffered saline (PBS), and then the brains were removed. For western blot experiments, the brains were stored at −70 °C and used. For immunohistochemistry, after perfusion with PBS followed by additional perfusion with 4% paraformaldehyde (PFA), the brain was removed and fixed overnight in 4% PFA, followed by successive fixation in 30% sucrose solution. Subsequently, they were successively sliced into 40μm coronal sections on a cryostat.

### 4.7. Immunohistochemistry and Immunofluorescence

#### 4.7.1. Immunohistochemistry

Immunohistochemistry for the visualization of neuronal nuclear antigen (NeuN), rat endothelial cells antigen 1 (RECA-1), and Iba-1 positive cells expression was performed. To begin the procedure, six sections on average were selected in each brain region, after being blocked with 10% normal goat and rabbit serum for 1 h. In brief, the sections were incubated overnight at 4 °C with a NeuN antibody (1:1000, ab134014, Abcam, Cambridge, UK), RECA-1 antibody (1:1000, ab9774, Abcam, Cambridge, UK), Iba-1 antibody (1:500; ab5079, Abcam, Abcam, Cambridge, UK), and 4HNE (1:200; HNE11-S, Alpha Diagnostic International, San Antonio, TX, USA). The sections were then incubated with the biotinylated rat and mouse secondary antibody (1:200; Vector Laboratories, Burlingame, CA, USA) with 0.3% Triton X-100 in PBS for 2 h at RT and subsequently incubated with antibody–biotin–avidin–peroxidase complex solution (Vector Elite ABC kit^®^; Vector Laboratories, Burlingame, CA, USA) for 1 h at RT. Finally, the sections were stained with a 3.3′-diaminobenzidine tetrahydrochloride (DAB kit; Vector Laboratories, Burlingame, CA, USA). The sections were finally mounted onto gelatin-coated slides. The slides were air-dried overnight at room temperature, and the coverslips were mounted using Permount^®^ (Vector Laboratories, Burlingame, CA, USA).

#### 4.7.2. Immunofluorescence

For immunofluorescence, after blocking (1× PBS, 2% normal donkey and goat serum and 0.3% Triton X-100) for 1 h, the sections were incubated overnight at 4  °C with a mixture of two of the following primary antibodies: mouse monoclonal antibody to RECA-1 antibody (1:1000, ab9774, Abcam, Cambridge, UK), Nrf2 (1:1000, ab31163, Abcam, Cambridge, UK), and rabbit monoclonal antibody to PDGFRβ (1:500, ab32570, Abcam, Cambridge, UK). The sections were then incubated with a mixture of Alexa Fluor 488-conjugated donkey anti-rabbit IgG and Alexa Fluor 594-conjugated goat anti-mouse IgG (1:1000; Molecular Probes, Eugene, OR) for 2 h at RT. Nuclei were visualized with 4′, 6-diamidino-2-phenylindole (DAPI). Slides were photographed for anti-goat Alexa Fluor 594 (red) and anti-mouse Alexa Fluor 488 (green) fluorescence with a confocal microscopy (Zeiss LSM 700; Zeiss; Oberkochen, Germany). For quantitative analysis of immunofluorescence data, the area of tissue fluorescence was analyzed using Image-Pro^®^ Plus software (Media Cybernetics, Silver Spring, MD, USA).

### 4.8. Western Blot

Protein was extracted from the rats’ hippocampus tissues by radioimmunoprecipitation assay (RIPA) buffer (ThermoFisher Scientific, Waltham, MA, USA). Protein (25 µg) was fractionated by 8–10% SDS-PAGE gel and transferred to a PVDF. After incubation with 5% nonfat milk in TBST (10 mM Tris, pH 7.6, 150 mM NaCl, 0.1% Tween 20) for 1 h at room temperature, they were incubated with anti-laminin (1:1000, ab11575, Abcam, Cambridge, UK), PECAM-1 (1:1200, LS-C348736, LSBio, LifeSpan BioSciences, Inc., Seattle, WA, USA), PDGFRβ (Abcam, 32570, 1:500), Nrf2 (1:1000, ab31163, Abcam, Cambridge, UK), Keap1 (1:1500, ab118285, Abcam, Cambridge, UK), GSTO1/2 (1:1000, SC-98560, Santa Cruz, CA, USA), SOD-1 (1:1000, SC-11407, Santa Cruz, CA, USA), and β-actin (1:100,000, SC-47778, Santa Cruz, CA, USA) overnight at 4 °C. Membranes were incubated with a 1:5000 dilution of horseradish peroxidase-conjugated anti-rabbit, mouse, goat secondary antibodies for 2 h. Blots were developed with enhanced chemiluminescence (Clarity™ Western ECL Substrate, Bio-Rad, Hercules, CA, USA). To quantitate, the protein band was performed with Image J software (U.S. National Institutes of Health, Bethesda, MD, USA).

### 4.9. Statistical Analysis

All data were acquired with at least three replicates and expressed as mean ± SEM. A statistical package for SPSS version 25.0 was used (IBM SPSS, Chicago, IL, USA) to compare the differences between groups (One-way ANOVA). Post hoc tests were Scheffe for multiple comparisons.

## Figures and Tables

**Figure 1 ijms-22-06285-f001:**
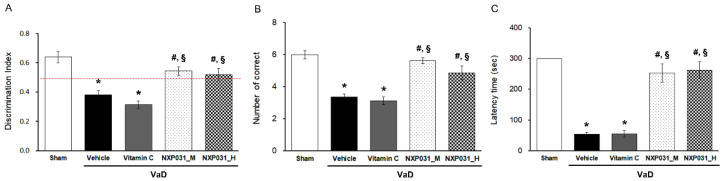
Effect of NXP031 on CCH-induced cognitive impairment using the NOR, radial 8-arm maze, and passive avoidance tests. (**A**) In the NOR test, rats treated with NXP031 showed significantly improved discrimination index (DI) compared to vehicle-treated rats. (**B**) In the radial 8-arm maze test, rats treated with NXP031 showed a significantly improved number of correct choices compared to vehicle-treated rats. (**C**) In the passive avoidance test, rats treated with NXP031 showed significantly increased latency times of suppression of behavior quantified on the basis of freezing compared to vehicle-treated rats. The data are presented as the mean ± S.E.M. * *p* < 0.05, compared with the sham group. # *p* < 0.05, compared with the VaD group. § *p* < 0.05, compared with the VaD + vitamin C group.

**Figure 2 ijms-22-06285-f002:**
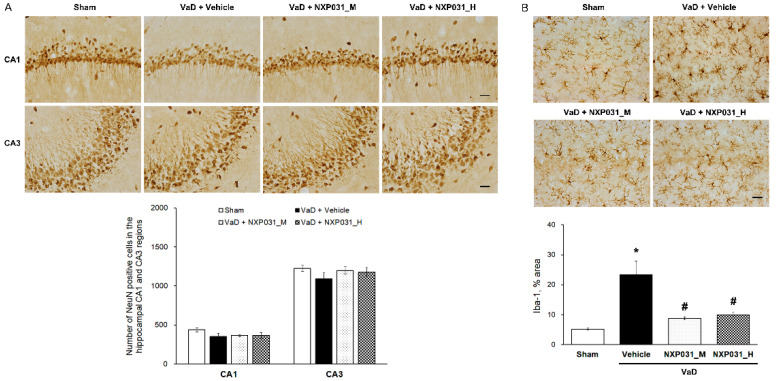
Effect of NXP031 on CCH-induced microglial activation in the hippocampus. (**A**) Representative photographs of immunostaining of NeuN-positive cells in the hippocampal CA1 and CA3 regions after CCH. (**B**) Representative photographs of immunostaining of Iba-1 in the hippocampal CA1 region after CCH. The data are presented as the mean ± S.E.M. * *p* < 0.05, compared with the sham group. # *p* < 0.05, compared with the VaD group. Scale bar: 50 μm.

**Figure 3 ijms-22-06285-f003:**
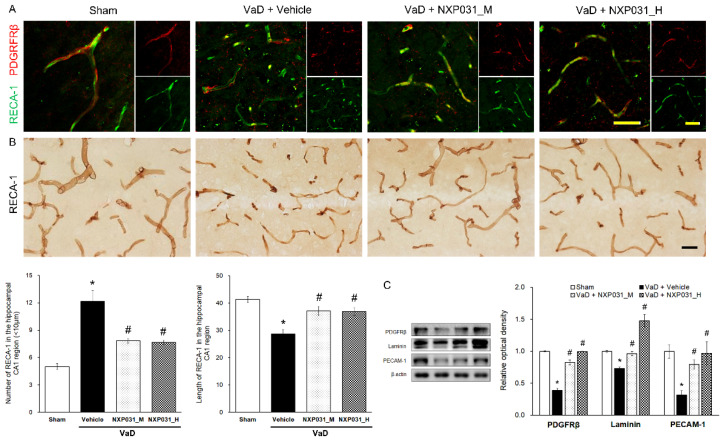
Effect of NXP031 on CCH-induced microvessels damage and BBB disruption in the hippocampus. (**A**) Representative photographs of immunofluorescence staining of PDGFRβ (red) and RECA-1 (green) in the hippocampal CA1 region after CCH. (**B**) Representative photographs of immunostaining of RECA-1 in the hippocampal CA1 region after CCH. NXP031 reduced the number of microvessels < 10 μm in the hippocampus. (**C**) Representative bands of PDGFRβ, Laminin, PECAM-1 protein expression in the hippocampus after CCH. The data are presented as the mean ± S.E.M. * *p* < 0.05 compared with the sham group. # *p* < 0.05, compared with the VaD group. Scale bar: 50 μm.

**Figure 4 ijms-22-06285-f004:**
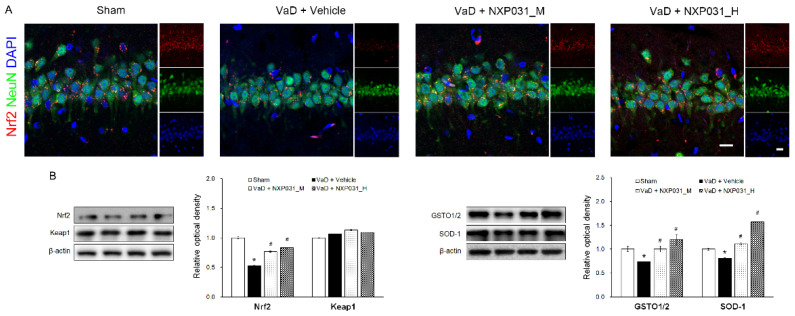
Effect of NXP031 on CCH-induced Nrf2 dysfunction in the hippocampus. (**A**) Representative photographs of immunofluorescence staining of Nrf2 (red) and NeuN (green)-positive cells in the hippocampal CA1 region after CCH. (**B**) Representative bands of Nrf2, keap1, GSTO1/2, and SOD-1 protein expression in the hippocampus after CCH. The data are presented as the mean ± S.E.M. * *p* < 0.05 compared with the sham group. # *p* < 0.05, compared with the VaD group. Scale bar: 20 μm.

**Figure 5 ijms-22-06285-f005:**
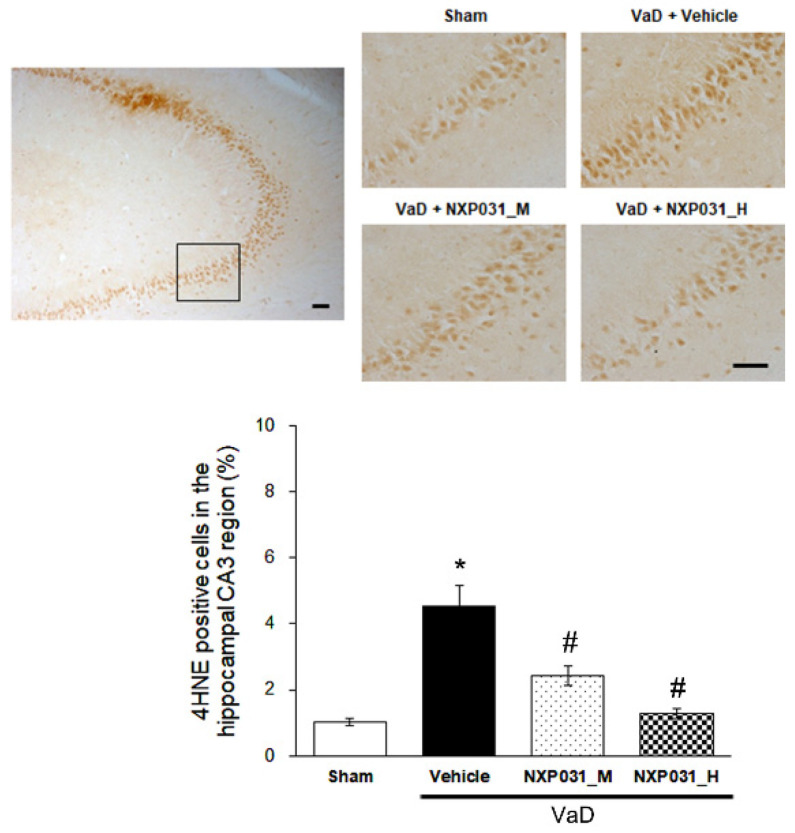
Effect of NXP031 on CCH-induced 4HNE production in the hippocampal CA3 region. Representative photographs of immunostaining of 4HNE expression in the hippocampal CA3 regions after CCH. The data are presented as the mean ± S.E.M. * *p* < 0.05, compared with the sham group. # *p* < 0.05, compared with the VaD group. Scale bar: 100 μm.

## Data Availability

Not applicable.

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
