# Peer review of "NXP031 Improves Cognitive Impairment in a Chronic Cerebral Hypoperfusion-Induced Vascular Dementia Rat Model through Nrf2 Signaling"

_ijms, 2021, doi:10.3390/ijms22126285_

Round 1

Reviewer 1 Report

In this manuscript, Lee Et al. describe their observations regarding NXP031, a novel DNA Aptamer/Vitamin C complex. NXP031 is hypothesized to protect against CCH-induced cognitive impairment and BBB disruption by enhancing antioxidant enzyme activity and lowering CCH-induced ROS production through potentiation of the Nrf2-ARE pathway in an animal model of VaD. The authors conduct a series of in-vivo experiments in which they induce VaD in Sprague-Dawley rats with BCCAO, treat them with NXP031 and then look for markers of cognitive impairment, microglial activation, microvessel damage, BBB disruption, oxidative stress response, and lipid peroxidation. The data indicate that NXP031 administration alleviates cognitive impairment, inhibits microglial activation in the hippocampus, reduces microvessel damage, BBB disruption, upregulates Nrf2 expression in the hippocampus, and suppresses 4HNE expression in the hippocampal regions. The authors interpret these data as supportive of their hypothesis.

The experiments described in the manuscript are straightforward. The test compound appears to improve behavioral attributes and physiological markers of cognitive impairment under in-vivo experimental conditions. The impact of such findings has particular relevance as a potential therapeutic option for cognitive impairment. However, questions concerning the following minor issues remain -

  1. Aptamer stability and interaction - The authors have not discussed several aspects related to the aptamer complex such as - aptamer degradation,  duration of action, interactions with intracellular targets, and cross-reactivity. What is the stability profile of NXP031? What is its half-life?
  2. Bioavailability in Brain - The authors have not provided data to show the BBB permeability and bioavailability of NXP031 in the brain? What amount of NXP031 administered intraperitoneally makes its way to the brain?

Author Response

Q1. Aptamer stability and interaction - The authors have not discussed several aspects related to the aptamer complex such as - aptamer degradation, duration of action, interactions with intracellular targets, and cross-reactivity. What is the stability profile of NXP031? What is its half-life?

Response : We genuinely appreciate all comments of reviewer #1. Actually, we performed the first study to demonstrate the protective effect of NXP031 on cognitive impairment and BBB disruption in a CCH-induced VaD model. We concentrated on the effect of NXP031 in the behavioral and neurobiological aspects, the description of the aptamer complex was insufficient in this manuscript. The antioxidant efficacy, aptamer degradation, duration of action, and interactions of NXP031 itself were detailed in previously published papers [54, 55], and this part was added in the discussion section. Summarizing previous studies, to confirm the ability of the aptamer to slow the oxidation rate of Vitamin C, Candidate aptamers were screened using CuSO4 as the oxidizing agent through SELEX and bioinformatics analysis. In addition, oxidation-reduction was analyzed using TEMPOL, an industrial oxidizing agent related to the nervous system. Aptamer increased the half-life of vitamin C by 3.3-fold in assay buffer and increased the half-life of vitamin C by 1.7-fold in Vitamin water® [54].

Degradation of vitamin C in Assay Buffer and vitaminwater® with and without Aptamer12

Choi et al. demonstrated that aptamin C has high binding affinity for vitamin C. Binding of Aptamin C with vitamin C was determined using ITC analysis. From binding isotherm, the enthalpy (ΔH), entropy (ΔS), and stoichiometry (n) of the binding reaction can be obtained.  The ITC measurements reveal that the change in entropy upon association with the Aptamin C is a major driving force for vitamin C. To prevent oxidation of vitamin C in the liquid state, all solutions were prepared with deionized water treated with nitrogen. The fluorescence was expressed and quantitatively analyzed when OPDA (o‐phenylenediamine) bound to DHA generated by oxidation of vitamin C. The degree of oxidation of vitamin C according to the concentration of Aptamin C (125, 250, 500 and 1000 nmol/L) was compared and confirmed by OPDA assay. The results demonstrated that the vitamin C conversion to dehydroascorbic acid was slower when the concentration of Aptamin C was higher. (Figure 2B). These data prove that Aptamin C is a dose‐dependent inhibitor of vitamin C oxidation.

Fig. 2 Inhibition of vitamin C oxidation by Aptamin C. A, Isothermal calorimetry. ITC of Aptamin Cb, Cf, and Ck binding to vitamin C. The binding affinity of Aptamin Cb is 2.13 µmol/L, Aptamin Cf is 0.89 µmol/L, and Aptamin Ck is 0.90 µmol/L. B, Comparison of OPDA assay according to the concentration of Aptamin C. Aptamin C concentrations are represented as ● (0 nmol/L), ■ (125 nmol/L), ▲ (250 nmol/L), ▼ (500 nmol/L), or ◆ (1000 nmol/L). C, Aptamin C prevents vitamin C oxidation for a long period of time.

We added the details and reference of the discussion section. The text was changed as follows (page 8, line 278-282):

Etc. (Please check the manuscript, we highlighted the changes in red)

Vitamin C is a primary antioxidant that mediates several beneficial effects on the immune system, inflammatory aging, redox pathways, and mitochondrial pathways [51]. Many mammalian species synthesize it from glucose in the liver; humans only absorb it through their diet [52]. The physiologically beneficial action of vitamin C comes from its ability to donate electrons as a reducing agent. In its solid form, vitamin C is stable, but it quickly decomposes when dissolved in water. It oxidizes easily by external factors such as air, moisture, or light [53]. A DNA aptamer that prevents oxidation of vitamin C and maximizes its effectiveness increased the half-life of vitamin C by 1.7-fold in certain commercially available vitamin water formulations [54]. In addition, vitamin C reduced oxidation to less than half-life of 2 weeks, but this DNA aptamer/Vitamin C complex maintained high binding affinity for an extended period (8 weeks) [55]. In addition to previous studies reporting neuroprotective effects in PD models [16], our research results demonstrate that the cerebrovascular integrity strengthening and neuroprotective effect of NXP031 protect from CCH-induced oxidative stress and cognitive deficit. This mechanism may strengthen the antioxidant defense system by activating the Nrf2-ARE signaling pathway, thereby maintaining the BBB’s integrity. This study suggests that the novel neuroprotective NXP031 is a promising treatment against cognitive impairments such as MCI and VaD.  

[54] Chiu, A.S.; Sankarapani, V.; Drabek, R.; Jackson, G.W.; Batchelor, R.H.; Kim, Y. Inhibition of vitamin C oxidation by DNA aptamers. Aptamers 2018, 2, 1–20.

[55] Choi, S.; Han, J.; Kim, J.H.; Kim, A.R.; Kim, S.H.; Lee, W.; Yoon, M.Y.; Kim, G.; Kim, Y.S. Advances in dermatology using DNA aptamer “Aptamin C” innovation: Oxidative stress prevention and effect maximization of vitamin C through antioxidation. J. Cosmet. Dermatol. 2020, 19, 970–976, doi:10.1111/jocd.13081.

Q2. Bioavailability in Brain - The authors have not provided data to show the BBB permeability and bioavailability of NXP031 in the brain? What amount of NXP031 administered intraperitoneally makes its way to the brain?

Response: We sincerely appreciate comments from reviewers. Cerebral blood vessels are made up of unique properties that form the blood-brain barrier (BBB). Endothelial cells that form blood vessels in all BBB components (endothelial, mural and glial cells, astrocytes and macrophages) play an important role in the proper functioning of the BBB [1]. Cerebrovascular damage, such as reduced microvessels density, leads to BBB destruction, and increased ROS contributes to endothelial dysfunction and increased BBB permeability [2]. It has been reported that the neuroprotective effects of natural antioxidants such as curcumin and resveratrol are related to vasodilatory function [3]. A previous study showed that intraperitoneal injection of resveratrol (known as a representative antioxidant) for 20 days maintains the BBB integrity in autoimmune encephalitis mice [4]. They assessed BBB leakage using Evans blue dye, and integrity was confirmed by tight junction protein analysis. Although we did not directly measure BBB permeability, we evaluated the BBB component proteins (PECAM-1, RECA-1, Laminin, PDGFRβ). In particular, we believe that our study proved BBB integrity by directly measuring the fragmentation of brain microvessels (RECA-1). However, the authors fully agree with the reviewer's advice and are contemplating the measurement of BBB permeability and bioavailability of NXP031 in brains in future studies.

[1] Daneman, R.; Prat, A. The blood–brain barrier. Cold Spring Harbor perspectives in biology. 20157, a020412. 1. doi: 10.1101/cshperspect.a020412.

[2] Enciu, A.M.; Gherghiceanu, M.; Popescu, B.O. Triggers and effectors of oxidative stress at blood-brain barrier level: Relevance for brain ageing and neurodegeneration. Oxid. Med. Cell. Longev. 2013,12, doi:10.1155/2013/297512.

[3] Mazzanti, G.; Di Giacomo, S. Curcumin and resveratrol in the management of cognitive disorders: What is the clinical evidence? Molecules 2016, 21, 1–27, doi:10.3390/molecules21091243.

[4] Wang, D.; Li, S.P.; Fu, J.S.; Zhang, S.; Bai, L.; Guo, L. Resveratrol defends blood-brain barrier integrity in experimental autoimmune encephalomyelitis mice. J. Neurophysiol. 2016, 116, 2173–2179, doi:10.1152/jn.00510.2016.

Reviewer 2 Report

In the present study, the authors investigate the neuroprotective effects of a new vitamin C/DNA aptamer complex (NXP031) against vascular dementia induced in a rat model by permanent bilateral common carotid artery occlusion. They show the beneficial effects of this new complex NXP031 in preventing chronic cerebral hypoperfusion-induced cognitive impairment, microglial activation, and oxidative. Although the manuscript is interesting, there are several concerns that should be addressed. Listed below some specific comments.

  • Results: The authors should define NXP031_M and _H at first mention in result section.

  • Due to a lack of fluency in the text, methods section and some parts in the manuscript need to be rewritten.

  • Figures are completely not sharp. Please enhance dpi.

  • Methods: It is not clear the protocol used for some behavioural test. Did you previously publish the modified (water as reward) radial 8 arm maze task for one of your studies? In addition to that, reference memory is a long-term memory and radial 8 arm maze require several days (at least 8) or weeks to train rats. How the authors could distinguish between reference and working memory errors with this modified protocol lasting for 24 hours? For such a reason, I would refrain to use “reference memory”: page 3, line 116.

  • Please correct the typos in figure 2 (NeuN graph).

  • Methods: The effects of ether on oxidative stress has been widely reported [DOI: 1016/0300-483x(93)90075-4; DOI: 10.1016/0006-2952(93)90171-r]. Please discuss.

  • A scheme that visualizes the presented major findings should be included.

Author Response

Q1. Results: The authors should define NXP031_M and _H at first mention in result section.

Response: First of all, we thank you for your meticulous review. According to reviewer’s advice, we added sentence in result section. The text was changed as follows (page 2, line 86-88.):

Etc. (Please check the manuscript, we highlighted the changes in red)

  1. Results

2.1. NXP031 alleviates CCH-induced cognitive impairment

We evaluated the effect of NXP031 on CCH-induced cognitive impairment using the novel object recognition (NOR), radial 8-arm maze, and passive avoidance tests. To confirm the difference between NXP031 concentrations, the mixture was mixed in two combinations. In this experiment, 200mg/4mg/kg of Ascorbic acid: Aptamin was used as a medium concentration, and 200mg/20mg/kg was used as a high concentration. The cognitive function tests showed that the NXP031 group had better spatial learning, recognition function, and short-term memory than the VaD (vehicle) group. Figure 1A presents the NOR test results. The NOR allows verifying cognitive function by assessing habituation, exploration, and new recognition periods, assessed as the difference in search times for new or familiar objects. The VaD group had a lower discrimination index (DI) in the NOR test than the Sham group. However, the VaD + NXP031_M and _H groups had a significantly higher DI than the VaD group (F = 25.34, p < 0.001). Figure 1B shows the radial 8-arm maze test results. Both of VaD + NXP031 groups had a significantly higher number of correct choices than the VaD group, showing a better spatial learning and memory function (F = 22.02, p < 0.001). In the passive avoidance test (which assesses short-term memory), the VaD + NXP031 groups also had significantly better results than the VaD group (Figure 1C) (F = 38.88, p < 0.001). And there is no difference between NXP031_M and _H groups. Overall, these results indicated that NXP031 attenuated CCH-induced cognitive impairment. However, Vitamin C administered alone did not affect cognitive function, so we excluded the Vitamin C group in all subsequent experiments.

Q2. Due to a lack of fluency in the text, methods section and some parts in the manuscript need to be rewritten.

Response: According to reviewer’s advice, we received English editing. We attach the certificate. The text was changed as follows (page 8, line 288-411.):

Etc. (Please check the manuscript, we highlighted the changes in red)

  1. Materials and Methods

4.1. Animals

Eight-week-old male Wistar rats (180 ± 20 g) were purchased for the present study (Orient Bio, Seongnam, Gyeonggi-do, Republic of Korea). Animals were housed in a place where water and food were freely consumed, and the environment was regulated (22 ± 2 °C, the humidity of 50%), and day and night cycle (12 h light / 12 h dark) controlled conditions. This study was conducted experiments by the National Institutes of Health Guide for the Care and Use of Laboratory Animals and approval of the Institutional Animal Care and Use Committee (IACUC) of the Kyung Hee University (KHSASP-19-080).

4.2. Bilateral common carotid arteries occlusion (BCCAO) procedure

Prior to surgery, the rats (body weight 250–350 g, 12 weeks old) were anesthetized with 2% isoflurane in 70% N2O and a balance of O2. The bilateral common carotid artery occlusions were performed to avoid damage to the vagus nerve and surrounding tissues. Both carotid arteries were exposed by a ventral midline incision and double-ligated with 3-0 silk (Ailee, Korea) immediately below the carotid bifurcation. Sham animals underwent the same operation procedure without vessel ligation.

4.3. NXP031 preparation

The aptamer is a single-stranded DNA that can selectively bind to a target molecule and is isolated by the SELEX method [54] and was provided by Nexmos Co., Ltd. The purified DNA aptamer was dissolved in 1mM MgCl2 PBS at 95 ℃ for 5 minutes and then cooled to room temperature to form a tertiary structure. L-Ascorbic acid (Sigma-Aldrich, MO, USA) was added at a ratio of 1:50 (w/w) and then adjusted to pH 5.7 to form the final NXP031. To confirm the difference between NXP031 concentrations, the mixture was mixed in two combinations. In this experiment, 200mg/4mg/kg of Ascorbic acid: Aptamin was used as a medium concentration, and 200mg/20mg/kg was used as a high concentration.

4.4. Experimental design

Male Wistar rats (12 weeks of age) were randomly divided into four groups (n = 10 in each group): The Sham group (vehicle), the VaD group (vehicle), VaD + NXP031 medium concentration (NXP031_M), and VaD + high concentration (NXP031_H). BCCAO procedure was performed at 12 weeks of age. 12 weeks after BCCAO, a radial 8-arm maze test was performed to confirm cognitive impairment induced by CCH. NXP031 groups were given treatment for 2 weeks simultaneously every day. Cognitive behavioral experiments were conducted for 2 weeks after treatment was terminated. All rats were sacrificed the next day after the end of all behavioral tests.

4.5. Behavior tests

4.5.1. Novel object recognition test

12 weeks after BCCAO, the novel object recognition (NOR) test is used to evaluate cognitive function in animals. On the day before the NOR test, rats were allowed to explore the empty box (60 × 60 × 40 cm) freely for 10 minutes. On the first day of the test, two identical objects (TF and TF) were placed in the upper right and lower left quadrants of the box, and rats were allowed to explore the objects in the box for 10 minutes. On the second day of the test, two different objects (TN and TF) are placed, where object F is the same object used in the familiarization session, and object N (with TN being dissimilar to TF) is the new object. The rats were then put back into the box for 10 minutes period of exploration. The behavior of rats was video recorded and the exploration of rats was analyzed to measure the time spent in exploring each object. Interest was based on the distance ≤ 1cm between the nose and the object. The evaluation of NOR memory was expressed as a percentage of the discrimination index (DI) calculated according to the following formula:

Discrimination Index =

TN

TN + TF

TN: time exploring novel object

TF: time exploring familiar object

4.5.2. Radial 8-arm maze test

The radial 8-arm maze test was performed to measure working memory. The radial 8-arm maze consisted of eight equally spaced arms radiating from a central octagonal platform. The rats were placed on the central octagonal platform, following which they were required to collect hidden water placed at the ends of each arms during a period of 8 minutes. Since there is water at the end of each arm, its contents are not visible from the central platform. An arm choice was scored if the rat crosses the halfway line with all four paws down an arm. Rats were allowed to enter arms in any order until all arms had been visited or 8 minutes had elapsed. Each arm entry was recorded, and re-entry into previously visited arms was recorded as an error. We calculated the number of correct choices made before the first error. The test was terminated either when the rat had entered all the arms or after 8 minutes. 

4.5.3. Passive avoidance test

The passive avoidance test is used to evaluate learning and memory in rodent model. The passive avoidance test apparatus is divided into two zones, a bright and dark chamber with lighting, and the floor is made of wire mesh. Rats are allowed to acclimate for 60 seconds without turning on the lights in an illuminated chamber and then turn on the lights. As soon as the rat moves to the dark chamber, an electric shock is applied for 0.5 mA for 2 seconds. The memory test was conducted 24 hours after training session. The latency times for rat to enter the dark chamber were measured for 300 seconds.

Q3. Figures are completely not sharp. Please enhance dpi.

 Response: As reviewer’s suggestion, we enhanced the dpi of the figures (300dpi).

Q4. Methods: It is not clear the protocol used for some behavioural test. Did you previously publish the modified (water as reward) radial 8 arm maze task for one of your studies? In addition to that, reference memory is a long-term memory and radial 8 arm maze require several days (at least 8) or weeks to train rats. How the authors could distinguish between reference and working memory errors with this modified protocol lasting for 24 hours? For such a reason, I would refrain to use “reference memory”: page 3, line 116.

Response: We thank the reviewer for raising this excellent point. Our previous study also identified cognitive impairment using a radial 8-arm maze task [1-3]. We agree with the reviewers that some method is misguided. The radial eight-armed maze task is to identify spatial learning and memory impairments, not reference memory. For this reason, we have excluded the “reference memory” from the manuscript.

[1] Song, M.K.; Kim, Y.J.; Lee, J. min; Kim, Y.J. Neurovascular integrative effects of long-term environmental enrichment on chronic cerebral hypoperfusion rat model. Brain Res. Bull. 2020, 163, 160–169, doi:10.1016/j.brainresbull.2020.07.020.

[2] Lee, J.M.; Park, J.M.; Song, M.K.; Oh, Y.J.; Kim, C.J.; Kim, Y.J. The ameliorative effects of exercise on cognitive impairment and white matter injury from blood-brain barrier disruption induced by chronic cerebral hypoperfusion in adolescent rats. Neurosci. Lett. 2017, 638, 83–89, doi:10.1016/j.neulet.2016.12.018.

[3] Kim, S. E.; Ko, I. G.; Kim, B. K.; Shin, M. S.; Cho, S.; Kim, C. J.; Jee, Y. S. Treadmill exercise prevents aging-induced failure of memory through an increase in neurogenesis and suppression of apoptosis in rat hippocampus. Experimental gerontology. 2010, 45, 357-365. Doi:10.1016/j.exger.2010.02.005.

Q5. Please correct the typos in figure 2 (NeuN graph).

Response: According to reviewer’s advice, the typo in figure 2 (NeuN graph) has been corrected.

Q6. Methods: The effects of ether on oxidative stress has been widely reported [DOI: 1016/0300-483x(93)90075-4; DOI: 10.1016/0006-2952(93)90171-r]. Please discuss.

Response: We thank the reviewer for raising this point. The ether was used in the 1800s as an anesthetic for a long time. Recently, the use of ether as an anesthetic has become very rare. Nevertheless, studies using ether as an anesthetic in animal experiments are still often seen [1-3]. Several previous studies have reported systemic (hepatic and renal) oxidative damage of ether [4,5], the evidence for oxidative damage in the brain is still unclear. However, it has been reported that ether, like alcohol, suppresses the central nervous system through the antagonizes NMDA, and causes slight changes in metabolism of neurotransmitter [6-8]. In the present study, we used ethers as an anesthesia for sacrifice in all groups. We found that the oxidative stress was significantly increased in the VaD group compared to the Sham group, it is considered that the effect of the CCH is greater than the ether. However, on the reviewer's advice, another anesthetic will be used to replace the ether in the next experiment.

[1] Awooda, H. A.; Lutfi, M. F.; Sharara G. M.; Saeed, A. M. Oxidative/nitrosative stress in rats subjected to focal cerebral ischemia/reperfusion. International journal of health sciences. 20159, 17–24. doi.org/10.12816/0024679.

[2] Wan, C.; Wei, Y.; Ma, J.; Geng, X. Protective effects of scoparone against ischemia‑reperfusion‑induced myocardial injury. Molecular Medicine Reports. 2018, 18, 1752-1760. doi.org/10.3892/mmr.2018.912.

[3] Zu, G.; Zhou, T.; Che, N.; Zhang, X. Salvianolic Acid A Protects Against Oxidative Stress and Apoptosis Induced by Intestinal Ischemia-Reperfusion Injury Through Activation of Nrf2/HO-1 Pathways. Cellular physiology and biochemistry : international journal of experimental cellular physiology, biochemistry, and pharmacology. 201849, 2320–2332. doi.org/10.1159/000493833.

[4] Liu, P. T.; Ioannides, C.; Shavila, J.; Symons, A. M.; Parke, D. V. Effects of ether anaesthesia and fasting on various cytochromes P450 of rat liver and kidney. Biochemical pharmacology. 199345, 871–877. doi.org/10.1016/0006-2952(93)90171-r.

[5] Liu, P. T.; Kentish, P. A.; Symons, A. M.; Parke, D. V. The effects of ether anaesthesia on oxidative stress in rats--dose response. Toxicology.1993, 80, 37–49. doi.org/10.1016/0300-483x(93)90075-4.

[6] Schwarting, R.; Huston, J.P. Short-term effects of ether, equithesin and droperidol/fentanyl on catecholamine and indolamine metabolism in the brain of the rat. Neuropharmacology. 1987, 26, 457–461, doi:10.1016/0028-3908(87)90027-X.

[7] Center, S.H.R.T.S.; Agency, U.S.E.P. Provisional Peer Reviewed Toxicity Values for Aluminum. Environ. Prot. 2006.

[8] Hudspith, M.J. Glutamate: A role in normal brain function, anaesthesia, analgesia and CNS injury. Br. J. Anaesth. 1997, 78, 731–747, doi:10.1093/bja/78.6.731.

Q7. A scheme that visualizes the presented major findings should be included.

Response: According to reviewer’s advice, we added a scheme to visualize the major findings presented.

Round 2

Reviewer 2 Report

Thank You for the diligently and thoroughly revised version of your manuscript.